# Heterologous Expression and Characterization of Plant Wax Ester Producing Enzymes

**DOI:** 10.3390/metabo12070577

**Published:** 2022-06-22

**Authors:** Daolin Cheng, Ling Li, Ludmila Rizhsky, Priyanka Bhandary, Basil J. Nikolau

**Affiliations:** 1Roy J. Carver Department of Biochemistry, Biophysics and Molecular Biology, Iowa State University, Ames, IA 50011, USA; daolinch@gmail.com (D.C.); liling@biology.msstate.edu (L.L.); ludmilar@iastate.edu (L.R.); 2Center for Metabolic Biology, Iowa State University, Ames, IA 50011, USA; 3Department of Genetics, Development and Cell Biology, Iowa State University, Ames, IA 50011, USA; bhandary@iastate.edu; 4Biological Sciences, Mississippi State University, Mississippi State, MS 39762, USA

**Keywords:** wax esters, wax synthase, heterologous expression, localization, seeds, yeast, plants, Arabidopsis

## Abstract

Wax esters are widely distributed among microbes, plants, and mammals, and they serve protective and energy storage functions. Three classes of enzymes catalyze the reaction between a fatty acyl alcohol and a fatty acyl-CoA, generating wax esters. Multiple isozymes of two of these enzyme classes, the membrane-bound O-acyltransferase class of wax synthase (WS) and the bifunctional wax synthase/diacylglycerol acyl transferase (WSD), co-exist in plants. Although WSD enzymes are known to produce the wax esters of the plant cuticle, the functionality of plant WS enzymes is less well characterized. In this study, we investigated the phylogenetic relationships among the 12 WS and 11 WSD isozymes that occur in Arabidopsis, and established two in vivo heterologous expression systems, in the yeast *Saccharomyces cerevisiae* and in Arabidopsis seeds to investigate the catalytic abilities of the WS enzymes. These two refactored wax assembly chassis were used to demonstrate that WS isozymes show distinct differences in the types of esters that can be assembled. We also determined the cellular and subcellular localization of two Arabidopsis WS isozymes. Additionally, using publicly available Arabidopsis transcriptomics data, we identified the co-expression modules of the 12 Arabidopsis WS coding genes. Collectively, these analyses suggest that WS genes may function in cuticle assembly and in supporting novel photosynthetic function(s).

## 1. Introduction

Wax esters are lipid molecules consisting of a fatty alcohol and a fatty acid moiety, covalently linked via an ester bond. They are found in a wide variety of organisms including bacteria, algae, plants, and animals [1,2]. Although wax esters are common in nature, with the exception of a few organisms, their natural abundance is relatively low. Rich sources of wax esters include whale oil, 95% of which is composed of wax esters [3]. Specifically, spermaceti oil from the head of sperm whales was prized in cosmetics, candle making, lubricants, and pharmaceuticals [4]. Once whaling was internationally banned in the 1980s, an alternative source of these lipids was promoted from jojoba oil, which is the oil isolated from the seeds of the desert shrub, *Simmondsia chinensis* [5,6]. Another rich source of these molecules is beeswax [7], which has a nearly 5000 year history of human applications [8,9]. In microbial organisms, such as *Acinetobacter calcoaceticus*, wax esters are used as a chemical form for storing carbon and energy.

Smaller quantities of wax esters are associated with discrete cellular or tissue structures in a variety of different organisms. This includes, for example, the surface cuticular lipids of terrestrial plants [10], the cuticle of insects [11,12], and mammalian cerumen (i.e., ear wax) [13]. In plants, wax esters contribute to the cuticle, and together with other cuticular lipids they form a thin hydrophobic layer that covers the outermost surfaces of the aerial organs of terrestrial plants, creating a barrier between the plant and the environment. The cuticle is of particular importance in controlling non-stomatal water loss and gas exchange [14]. In addition, the cuticle has an important function in preventing organ fusion during plant development [15,16], in pollen–stigma interactions [17], and mediating interactions between plants and microbial colonizers of plant surfaces [18,19].

There are quantitative and qualitative differences in the chemistry of plant cuticular wax esters isolated from different species. Even within a single species there are organ-specific variations among different cuticles. For example, the Arabidopsis leaf and stem cuticle contains only about 0.2% and up to 3% of wax esters, respectively [20], whereas 85% of the leaf cuticle of the carnauba palm (*Copernicia cerifera*) consists of wax esters [2]. There are also considerable chemical variations in these waxes, consisting of different alkyl chain lengths of the fatty acid and fatty alcohol moieties [21]. These characteristics are probably a consequence of diversity in the biosynthetic machinery that generates these lipids.

The reaction of wax ester biosynthesis is the acylation of a fatty alcohol with a fatty acid, the latter being donated by an acyl-CoA substrate. Three unrelated families of enzymes catalyze this reaction. The first of these enzymes to be described was the wax synthase (WS) class of enzymes, isolated from jojoba seeds [22], and this enzyme belongs to the membrane-bound O-acyltransferase (MBOAT) family of enzymes [23]. Another wax ester producing family of enzymes is the bi-functional wax synthase/acyl-CoA:diacylglycerol acyl transferase (WSD), which was first isolated from the bacterium *A. calcoaceticus* [24,25]. Characterization of such an enzyme in Arabidopsis has identified the *WSD1* locus as being responsible for the production of the wax esters of the plant cuticle [26]. Further analyses of the 11 WSD-like isozymes that occur in Arabidopsis have identified their role in producing wax esters and as being important in establishing a barrier that contributes to drought tolerance in plants [27]. The third wax ester forming enzyme family, is the mammalian wax synthase, first identified in mice [28], but no obvious plant homologs have been described or can be found in plant genome sequences.

In this study we queried the differences between the WS and the WSD type enzymes of Arabidopsis and developed two heterologous expression systems to specifically characterize the catalytic capabilities of WSs relative to the ability to assemble wax esters. One of these expression platforms utilize the yeast, *Saccharomyces cerevisiae*, and the second utilizes transgenic Arabidopsis seeds. The yeast platform can be externally fed different alcohols and fatty acids to ascertain the in vivo ability of the individual gene-products to catalyze the assembly of wax esters. In the Arabidopsis seed platform, we co-expressed the putative wax ester-forming enzyme with the jojoba fatty acyl-CoA reductase [22], which generates the fatty alcohol substrate required for assembly of the wax ester. These platforms were complementary in their ability to generate functional wax ester producing enzymes and provided insights to WS enzymes with distinct substrate specificities. Additional characterizations included the visualization of the tissue and subcellular localizations of WS isozymes by the use of GUS/GFP reporter transgenes. Finally, we also determined and compared the co-expression gene network associated with the Arabidopsis WS and WSD genes. Collectively, these characterizations suggest that the WS genes may have physiological functions that are distinct from those of the WSD genes.

## 2. Results

### 2.1. Sequence Characterization and Phylogenetic Classification of WS and WSD Proteins

Characteristics of the 12 MBOAT-type WS homologs and the 11 WSD homologs that occur in the Arabidopsis genome are identified in Table 1. The WS proteins share ~50% primary sequence identity with each other and with the archetypal jojoba WS sequence [22]. The 11 WSD homologs also share ~50% sequence identity with each other and the archetypal *A. calcoaceticus* WSD protein sequence [24], but there is insignificant homology between the WS and WSD homologs. The WS-like proteins range between 333 and 351 residues, and hydropathy analysis [29] of these sequences predicts that they are membrane associated, containing seven or eight transmembrane spanning domains, which is typical of the MBOAT-type proteins [23]. In contrast, the WSD proteins are larger (ranging between 479 and 522 residues) and hydropathy analysis predicts that they have a single transmembrane spanning domain. 

Figure 1 shows the phylogenetic relationship of a broader collection of 375 MBOAT-type WS homologs (listed in Appendix A) found by BLASTP query of the NCBI RefSeq sequence database [30] using the jojoba WS protein [22] as the query (bit score > 200). This analysis identified six major clades (WS-A to WS-F). All of these proteins are sourced from the Eudicot phylum of the Plantae kingdom, and they belong either to the Asterids or Rosids phylogenetic Clades. Four of the clades (WS-C to WS-F) are populated by proteins that belong to both the Asterids and Rosids clades, encompassing 14 Orders, and these are interspersed among all four clades. In contrast, the WS-A clade contains proteins that belong only to the Asterids Clade, and specifically the Family Brassicaceae, whereas the WS-B proteins are from ten different Orders, within the Rosids (Appendix A). 

Similar BLASTP analyses the NCBI RefSeq database using the archetypal WSD protein sequence from *A. calcoaceticus* (ADP1) [31] and the Arabidopsis AtWSD1 sequence [26] as the initial queries, identified 1054 WSD homologs (bit score > 200); these sequences are identified in Appendix A. Phylogenetic relationship among these sequences indicate that they segregate into five clades, labeled as WSD-A to WSD-E (Figure 2). In contrast, to the WS sequences that are restricted only to Plantae, the WSD sequences are found in Plantae and bacteria (Appendix A). The two clades that are most distinct from the query sequence (WSD-D and WSD-E) are populated by only bacterial sequences, which are restricted to Proteobacteria and Actinobacteria. The sequences populating WSD-A to WSD-C clades are only from Plantae. The Plantae sequences divide into the Eudicots (clade WSD-A) and Monocots (clade WSD-B), and clade WSD-C contains sequences that are from a mixture of vascular Eudicots and Monocots, and non-vascular Bryophytes (mosses). All 11 Arabidopsis WSD homologs reside in the WSD-A clade, within the Brassicaceae Family subclade.

### 2.2. Functional Characterization of the WS Proteins by Heterologous Expression in Yeast

Prior characterizations of the WSD proteins have identified their catalytic capabilities in assembling wax esters as components of the plant cuticle [26,27,32]. However, the catalytic capabilities of MBOAT-type WS proteins have only been demonstrated with the jojoba WS [22], but little is known of the capabilities of the diverse WS that occur uniquely in plants. Therefore, we selected three Arabidopsis WS sequences (At5g55320, At5g55340, and At5g55380) and developed two heterologous expression systems to characterize the ability of these proteins to assemble wax esters. These three proteins were selected because they are encoded by genes situated in the adjoining region of the Arabidopsis genome on Chromosome 5, which also contains an additional five sequentially arranged WS-like genes. Additionally, publicly available RNA-Seq data (Table 1 and Appendix A) indicates that the three selected WS genes show different spatial and temporal expression patterns from each other, with At5g55320 being specifically expressed in the reproductive organs (e.g., flowers, siliques, and developing seeds); At5g55340 shows a more diverse expression pattern among different tissues and organs; and At5g55380 is the most highly expressed gene from this phylogenetic clade [33]. The characterization of these three homologs could, therefore, reveal the functional diversity that may have arisen following the evolutionary gene duplication events that probably generated the collection of WS-gene family in this region of the Arabidopsis genome. Additionally, we selected WS-like sequences sourced from soybean (GenBank Accession ACU22750.1), maize (GenBank Accession NP_001147171.1, encoded by Zm00001d048476), and the moss, *Physcomitrella patens* (GenBank Accession XP_001765175). Functional analyses of these latter proteins could, thus, provide information about functional diversification among WS genes associated with different plant phyla. 

The yeast strain (H1246) that we used to express the putative wax ester-producing enzymes carries four knock-out mutant alleles at the *dga1*, *lro1*, *are1*, and *are2* loci; collectively these mutations eliminate neutral lipid biosynthesis [34] (i.e., TAG and sterol ester biosynthesis is eliminated in this host). This strain has proven ideal for testing the ability of novel gene products to assemble wax esters and other neutral lipids [35,36,37]. Each of the putative WS genes were expressed in this strain under the transcriptional control of the *GAL4* promoter. Moreover, each WS protein was expressed with a C-terminal His_6_-tag, which facilitated immunological detection of successful expression of these recombinant proteins. The resulting strains were initially evaluated by immunological Western blot analysis of protein extracts, using commercially available anti-His tag serum. This assay established that, among the six strains that were developed, expression of the Arabidopsis At5g55340, the *P. patens* (XP_001765175), and the maize (NP_001147171.1) WSs were immunologically detectable (Figure 3A). In contrast, no immunologically reactive proteins were detected in extracts from the control strain and all other recombinant strains.

The wax ester producing capabilities of the putative WS enzymes was demonstrated by feeding a variety of potential fatty acid and fatty alcohol substrates to the yeast strains expressing each of the plant WS sequences. In these experiments, fatty acids of different chain lengths and many potential fatty alcohol-substrates were included in the medium, including primary alcohols (e.g., methanol and ethanol), branched chain alcohols, sterols, and aromatic alcohols. One of the complexities that needed to be overcome in this series of experiments was the normal metabolism of yeast with exogenously provided fatty acyl-substrates. Namely, exogenously provided fatty alcohols can be oxidized to fatty acids, these fatty acids, and other exogenously fed fatty acids are metabolized either by elongation or ß-oxidation to longer and shorter chain lengths, respectively [38]. These confounding issues were overcome by feeding yeast strains with fatty acids and fatty alcohols of an odd number of carbon atoms, which are not normally produced by yeast. Therefore, because the endogenous fatty acid elongation and ß-oxidation systems can change the chain-lengths of these exogenously fed precursors by an even number of carbon atoms, they maintained their odd number chain length character, enabling their metabolic tracking into wax ester assembly. 

Lipid extracts prepared from these “fed” yeast strains were initially analyzed by TLC. Wax esters were not detectable in lipids extracted from the non-recombinant yeast strains or from the strains carrying the empty control vector, grown in the galactose-containing induction medium with or without supplemented fatty acids. In contrast, the inclusion of fatty acids and fatty alcohols in the cultivation medium of strains expressing WS genes led to the formation of wax esters in two of the six recombinant strains that were evaluated. Namely, the yeast strain expressing the Arabidopsis At5g55340 WS and the maize NP_001147171.1 WS (Figure 3B). The identity of the wax ester products was authenticated by GC-MS analysis, which showed that the yeast strain expressing the At5g55340 WS assembled wax esters when it was grown in the presence of both fatty acids and fatty alcohol mixtures of 13-, 15-, 17-, and 19-carbon chain lengths. By combinatorial feeding this yeast strain with combinations of a fatty acid (13:0, 15:0, 17:0, or 19:0) and the homologous fatty alcohol (C13:0, C15:0, C17:0, or C19:0), we determined that this enzyme prefers that the two substrates have near equal chain length with an apparent optimum at 13-carbon chain length for both the fatty acid and fatty alcohol moieties (Figure 3C). Similarly, by testing saturated and unsaturated substrates, we determined that this enzyme has a higher preference for monounsaturated substrates for both the fatty acid and fatty alcohol moieties (Figure 3D).

Similar characterizations of the maize WS (NP_001147171.1), indicates that this enzyme has a different substrate preference, assembling esters with medium-chain fatty acids, but it appears incapable of assembling esters with fatty alcohols of more than 15-carbon chain length (Figure 3E). Moreover, this WS prefers assembling esters with unsaturated moieties (Figure 3F). Another interesting feature of the maize WS is the ability to assemble esters with ethanol, benzyl alcohol, and phenylethyl alcohol. The preferred ethyl ester is that assembled with the unsaturated, 14:1 fatty acid (Figure 3G) and this fatty acyl preference is maintained in the assembly of benzyl esters, but yield is considerably lower (Figure 3H).

### 2.3. Functional Characterization of the WS Proteins by Heterologous Expression in Arabidopsis Seeds

Seven expression vectors were constructed for evaluating WS homologs *in planta*. These vectors also carried the jojoba 3-ketoacyl-CoA synthase (KCS) [22] and the jojoba fatty acid reductase (FAR) [39], and all three transgenes were under the transcriptional control of the soybean glycinin regulatory sequence, which guide the expression of the transgenes to developing embryos of transgenic plants [40,41]. Jojoba KCS was chosen because it can elongate fatty acyl-CoA to chain lengths of up to 24 carbons [42], and these acyl-CoAs can be chemically reduced by the jojoba FAR to generate primary long chain fatty alcohols in planta [39]. One plasmid (pKF) did not carry any WS-like ORFs and was used as a negative control. The other six plasmids, named pKFW1 to pKFW6, contain one of the WS ORFs that had also been characterized by expression in yeast. The recovered transgenic Arabidopsis plants were evaluated by PCR-based genotyping, which identified that all recovered hygromycin-resistant plants carried the appropriate WS transgene (Figure 4A). RT-PCR analysis evaluated the expression of each transgene, and as a control we evaluated the expression of the *Actin-2* mRNA (Figure 4A). The successful expression of each transgene was indicated by the ability of each RT-PCR assay to detect the appropriate mRNA, which was confirmed by the direct sequencing of each product. 

Lipids were extracted from mature transgenic T3 generation seeds, and TLC was used to analyze the lipid classes. Primuline-staining of the developed TLC plates identified the presence of a putative wax ester spot in the seed lipids extracted from transgenic lines expressing the Arabidopsis At5g55380 (pKFW3); this TLC-spot was absent from extracts prepared from the control transgenic plants (transformed with pKF). Additional analyses to authenticate this identification was conducted by GC/MS analysis of the extracts. Extracts from transgenic lines expressing only KCS and FAR (i.e., transformed with the plasmid, pKF) revealed the occurrence of novel GC peaks, which were absent from the non-transgenic control seeds, and these were identified as elongated fatty acids, 20:0 and 24:0, and monounsaturated 20-, 22-, and 24-carbon primary alcohols (Figure 4B). Moreover, analysis of lipid extracts prepared from the transgenic seeds that also expressed the At5g55380 WS, identified the occurrence of wax esters, consisting of 16- and 18- carbon fatty acids, esterified with these long chain monounsaturated primary alcohols of 20-, 22-, and 24-carbon chain lengths (Figure 4B). Wax esters were only detectable in extracts prepared from the transgenic seeds expressing the At5g55380-encoded WS and were undetectable in all other transgenic lines that were developed. Quantitative analysis of these data indicates that the co-expression of the jojoba KCS and FAR increases the yield of free fatty acids by approximately 3-fold and yields ~1% (*w*/*w*) of fatty alcohols (Figure 4C). The co-expression of the At5g55380-encoded WS with the jojoba KCS and FAR yields ~1.5% (*w*/*w*) of wax esters in the seeds. 

### 2.4. Expression Patterns of the Arabidopsis WS Genes

Prior characterizations have determined the differential expression patterns of the *WSD* genes in different tissues of Arabidopsis [27,43]. Herein, we broadened such characterizations by examining the expression patterns of the WS genes using transgenic Arabidopsis plants that carried promoter-GFP/GUS constructs, focusing these studies on the two specific genes, At5g55380 and At5g55340. These constructs were assembled by using DNA fragments containing 1330-bp upstream of the translational ATG-start site of At5g55340, or 1551-bp upstream of the translational ATG-start site of At5g55380 (Appendix A). The latter promoter fragment overlapped with the 3′-end of the adjoining gene (At5g55390); therefore we also made a construct that used only a 280-bp upstream promoter fragment that avoided this overlap. Because the expression pattern of the reporter gene that was generated by these two promoter fragments from At5g55380 are indistinguishable, we conclude that the 280-bp fragment is the effective minimal promoter that directs the expression of this WS gene. In addition to these promoter-reporter constructs, we investigated the subcellular localization of the two WS proteins by using constructs that fused the GFP-reporter gene at the C-terminal translational stop codon of each ORF (Appendix A).

Figure 5 compares the GUS expression patterns generated by the promoters of the WS genes, At5g55380 and At5g55340, in different organs of transgenic Arabidopsis seedlings, at different stages of growth. These data clearly visualize different developmental expression programs of each WS gene. For example, although At5g55380 is expressed in almost all organs of young seedling, i.e., at 9 days after imbibition (DAI), the expression of At5g55340 is restricted to the young true leaves (tl) and the shoot meristem (sm) region. Later in seedling development, as the floral bolt is beginning to emerge (at 19 DAI), At5g55380 shows strong expression in the vegetative organs, whereas expression of At5g55340 is primarily restricted to the shoot meristem region of the seedling. This differential expression between the two genes continues as the seedlings near maturity (at 40 DAI), with At5g55380 showing strong expression in almost all of the organs of the flowers (notably, no expression is detected in petals) and the developing siliques, whereas expression of At5g55340 is considerably weaker, and restricted to discrete tissues, being particularly strong in the receptacles (r). These two genes show differentially distinct tissue expression patterns in the roots, with At5g55340 expression being concentrated in the root cap (rcp) and the emerging lateral root tips (elr), while expression of At5g55380 is not detectable in these tissues, but rather concentrated in the root cortex (rc) and root vasculature (rv). Collectively, these data indicate that the expression of these two WS genes is complementary, with At5g55340 being expressed in tissues that do not show At5g55380 expression, and vice versa. 

Using the transgenic lines that carry the GFP gene fused at the C-terminus of each WS ORF, we characterized the subcellular localization of the WS proteins encoded by At5g55340 and At5g55380. Figure 6 compares the localization of the GFP fluorescence expressed from these transgenes, compared to the auto-fluorescence that identifies plastids (i.e., chloroplasts). Both WS proteins appear to guide the localization of the GFP reporter to extra-plastidial space of the cells. Based on the fact that these WS proteins are predicted to be integral membrane proteins, we surmise that the GFP fluorescence is associated with either the plasma membrane or cytosolically localized membrane systems, such as the endoplasmic reticulum. Such localizations are also consistent with computational predictions by PSORT [43] and SUBA [44], which indicate that both proteins are likely to be associated with the plasma membrane (Table 1).

### 2.5. Co-Expression Gene Network Associated with the Arabidopsis MBOAT-Like WS and WSD Coding Genes

Computational and statistical analysis of the publicly available Arabidopsis RNA-Seq transcriptome data provide a broader perspective on the expression patterns of all 23 Arabidopsis genes that code for wax ester producing enzymes. The data that we queried were collected from over 5200 individual RNA-Seq experiments that determined the Arabidopsis global transcriptome of different organs and tissues isolated from plants grown under a wide range of environmental conditions (Appendix A). These data were downloaded from NCBI Sequence Read Archive (www.ncbi.nlm.nih.gov/sra, accessed on 19 June 2022), and were explored using the computational platform, MetaOmGraph [45]; the outcomes of these analyses are summarized in Figure 7 and Table 1. Specifically, all 23 identified WS and WSD genes are transcriptionally active, but each gene has a distinct developmental profile that is differentially affected by environmental conditions. The violin plot (Figure 7A) indicates that the WS and WSD genes display levels of gene expression that range over five orders of magnitude from minimal to maximal levels of expression. The mean expression levels of each WS and WSD gene range over a 50-fold difference between the lowest and highest level of expression. Moreover, the box plots of these data (Appendix A) illustrate the range of expression by each individual wax ester producing gene, which indicates that the WSD-like gene, At3g49200, shows the lowest range of expression, and the highest range of expression is shown by the WSD-like gene, At5g12420. Table 1 identifies the organs where the maximal expression of each wax ester producing gene is detected and also identifies the maximal level of expression of each gene in that organ; there is a 50-fold range between the lowest and highest level of maximal expression. The majority of the organs where maximal expression occurs are aerial tissues, consistent with the expectation that as components of the cuticle, genes that are involved in producing wax esters would be highly expressed in aerial organs.

Figure 7B illustrates the expression correlations among each pair of WS and WSD genes. The strongest correlations are among two groups of four genes each; one group consists of four WSD genes (At1g72110, At5g37300, At2g38995, and At5g3390), and the other group consists of two WSD genes (At2g43255 and At3g49200) and two WS genes (At1g34490 and At1g34500). The other 16 WS and WSD genes that were evaluated show expression patterns with low correlations, indicative of relatively independent expression profiles.

Additional correlation analyses compared the expression pattern of each of the 12 WS genes and 11 WSD genes with the expression pattern of all the genes in the Arabidopsis genome (Appendix A). Many of the WS genes (i.e., At1g34490, At1g34500, At3g51970, At5g51420, At5g55320, At5g55330, At5g55340, At5g55350, At5g55360, At5g55370, and At5g55380) and WSD genes (i.e., At1g72110, At3g49200, At5g22490, and At5g53380) each show expression patterns that correlate with less than 60 other genes in the Arabidopsis genome; indeed, expression patterns of eight of these genes (i.e., At1g34500, At1g72110, At3g49200, At5g22490, At5g51420, At5g55320, At5g55350, At5g55360, and At5g55370) do not significantly correlate with the expression of any other gene in the Arabidopsis genome. Therefore, the majority (i.e., 14 of the 23) of the Arabidopsis WS and WSD genes show distinct expression patterns, relative to the other genes in the genome. 

The expression patterns of the other nine wax ester producing genes (eight WSD and one WS gene) collectively correlate (>0.6 Pearson’s correlation coefficient) with the expression of a list of ~4700 unique Arabidopsis genes (Appendix A). More significantly, and possibly indicative of the physiological functions of these wax ester forming genes, this non-redundant list of ~4700 genes that correlate with the expression of the nine wax ester forming genes can be separated into two distinct lists, which, for convenience, we call Group A and Group B (Appendix A). Group A consists of 1763 unique genes whose expressions correlate with the expression of the three WSD genes (At1g34520, At3g49190, and At5g12420) and the single WS gene (At1g34520). Group B consists of 2947 genes whose expressions correlate with the expression of five WSD genes (i.e., At5g53390, At5g37300, At5g16350, At2g38995, and At3g49210). 

These two gene lists were analyzed with the ShinyGO tool [46] to identify if they are associated with distinct biological processes that may provide insights on the physiological functions associated with the two sets of wax ester forming enzymes. The Group A and Group B gene lists identified distinct GO annotation terms, KEGG pathways and ShinyGO enrichment terms (Appendix A). These analyses illustrate that whereas both groups of genes are associated with metabolic processes, there are terms and pathways that distinguish the two groups. Specifically, Group A is enriched in terms such as “detoxification”, “development and differentiation”, “stress”, and “response”, which collectively indicate that these wax ester producing genes (the three WSD-coding genes At5g12420, At3g49190, and At1g34520, and the WS-coding gene At1g34520) are co-expressed with genes that may be involved in some type of stress response, possibly generated by establishing a barrier, consistent with the plant cuticle. In contrast, Group B annotations are enriched in terms and pathways that are associated with chloroplasts, chlorophyll, and photosynthesis, and thereby indicating that the five WSD coding genes, At5g53390, At5g37300, At5g16350, At2g38995, and At3g49210, may be associated with processes that support chloroplast molecular structure(s) associated with the processes of photosynthesis. 

## 3. Discussion

Wax esters are widely distributed among many phylogenetic taxa, occurring in animals, plants, and microorganisms [1,2]. Reflective of their chemical inertness, wax esters serve roles as either chemical forms of energy storage (e.g., jojoba seed oil and spermaceti oil), or as structural water barriers (e.g., cuticle components and beeswax). Wax esters are formed by the reaction between a fatty alcohol and a fatty acyl-CoA, and biological systems appear to have evolved three families of enzymes to catalyze this reaction. Two of these coexist in plants, the MBOAT-type WS and the bifunctional WSD-type enzymes. Extensive molecular genetic studies have characterized that some of the WSD-type enzymes are important in producing the wax ester components of the plant cuticle [26] and have a role in developing drought tolerance [27,32,47,48,49]. In contrast, even though WS was the first wax ester producing enzyme to be characterized from plants (from jojoba seeds) [22], these enzymes are less well characterized. 

In this study we focused on these less well characterized WS genes, a class of wax ester producing enzymes that appear to be restricted to Plantae. We developed two heterologous expression systems (one with yeast and the second with Arabidopsis seeds) for these WS enzymes and used these systems to evaluate the catalytic capabilities of WS proteins from Arabidopsis, and one each from maize, soybean, and a moss, *P.*
*patens*. Even though these two systems proved complementary, they were not completely successful in evaluating the catalytic capabilities of all the tested sequences. Thus, only four of the six evaluated sequences were successfully expressed in these systems, three in the yeast platform and one additional sequence in the seed platform. These experiments provided direct confirmation that the four tested WS proteins have the catalytic capability to assemble wax esters. Maybe more insightful was the fact that these four WS enzymes have distinct substrate preferences, and thus assemble different types of esters, including unsaturated wax esters, benzyl esters, and ethyl esters. These novel catalytic capabilities may be useful in the development of biobased systems for producing molecules that have applications as biofuels or as renewable replacements of petroleum derived lubricants and surfactants [50,51].

Although the novel wax esters produced by some of these enzymes may have utility in biobased industrial applications, the *in planta* functionality of these esters are not necessarily obvious. For example, the types of wax esters produced by the WS enzymes we evaluated are not constituents of the plant cuticle; thus these enzymes may not directly contribute to the assembly of this extracellular lipid barrier. These enzymes may, however, have roles in regulating cuticle deposition by affecting other components that directly produce the metabolites needed for the assembly of the cuticle [52]. Indeed, our cellular distribution and subcellular localization studies with GUS and GFP fusion transgenes indicate that the expression of two of the Arabidopsis WS genes is not limited to the epidermis, which is the cell layer that generates the cuticle. In contrast, the expression of the WSD1 enzyme that generates the wax esters of the cuticle is restricted to the epidermis [26]. 

The hypothesis that the Arabidopsis WS enzymes may not be generating the wax esters of the cuticle was further supported by the co-expression analysis of these genes in the context of all other genes in the Arabidopsis genome. Individually, the majority of the WS genes (10 out of 11) showed distinct expression patterns that correlated with less than 60 other genes of the Arabidopsis genome. Thus, although we are not able to find co-expression modules that may be suggestive of a physiological function of most Arabidopsis WS genes, the expression patterns of seven of the twelve WSD genes and one of the WS genes, correlated with the expression patterns of a significant number of other genes (~4700) in the Arabidopsis genome. Functional annotation analyses of these co-expressed modules indicate that these genes may be involved in two distinct processes, both of which are associated with metabolism. One of these is related to chloroplasts and photosynthesis, whereas the second is associated with establishing a barrier that provides protection from a stress, consistent with the prior finding that one of these genes (*WSD1*) supports the assembly of the cuticle, and many of the other WSD genes may provide protection from drought [27,32,47,48].

## 4. Materials and Methods

### 4.1. Phylogenetic Analysis

BLASTP analysis [53,54] was conducted to identify protein sequences in the NCBI NR database that are homologs to the MBOAT-class of WS proteins, using as query the experimentally characterized jojoba WS sequence (Accession AAD38041.1) [22]. The eleven members of the Arabidopsis WSD gene family had been previously defined [26,27]. These sequences were imported to MEGA 4.0 and they were pairwise aligned, and phylogenetic trees were constructed [55]. The out-group for the phylogenetic tree of the WS homologs was the experimentally characterized WSD sequences, and the out-group for the phylogenetic tree of the WSD homologs was the experimentally characterized MBOAT-class of WS sequences. Each phylogenetic tree was assembled with the neighbor joining method and their statistical rigor tested with a bootstrap calculation of *n* = 1000 [56].

### 4.2. Construction of Plant Expression Cassettes

Plant WS expression vectors were constructed with a derivative of a plasmid (pFWS3) provided by Dr. Edgar Cahoon (University of Nebraska, Lincoln, NE, USA) [57]. Originally this plasmid carried the plant marker gene, DsRed, and three ORFs whose transcription were under the control of the seed specific, *Glycinin* promoter [41]. These three ORFs encoded for the jojoba KCS, which is involved in the elongation of fatty acids, the jojoba FAR, which catalyzes formation of fatty alcohols from fatty acyl-CoAs, and the jojoba WS enzyme. This plasmid was modified by: (1) the replacement of the plant marker gene, DsRed, with a hygromycin resistance gene; and (2) the excision of the jojoba WS ORF. The resulting vector was named pKF, which was used to generate control transgenic plants. Further modifications of pKF were the individual cloning of 6 WS ORFs at the original location of the jojoba WS ORF. 

The organ- and tissue-specific expression patterns of WS genes were assessed using transgenic plants that carried a reporter gene (GUS) fused to different promoter elements from At5g55340 or At5g55380. Three different constructs were generated in which different promoter fragments were fused to the reporter gene, in the expression transformation vector, pBGWFS7 [58]. These promoter fragments were PCR-amplified from isolated genomic DNA, using primers that contained the *attB*-adapted sequence at the 5′-end of the primer followed by the WS-gene specific sequence, which facilitated subsequent cloning using Gateway™ Cloning Technology (Invitrogen, Carlsbad, CA, USA). These *attB*-adapted sequences were: (a) 5′-GGGGACAAGTTTGTACAAAAAAGCAGGCTTC, which was fused to each forward gene-specific primer; and (b) 5′-GGGGACCACTTTGTACAAGAAAGCTGGGTC, which was fused to each reverse gene-specific primer. For the At5g55380 WS gene, two promoter fragments were amplified, using forward gene-specific sequences: (a) 5′-ACCTTGTTCCTCCGCCACT-3′; or (b) 5′-ATCAGCACTCATTATCCTT-3′, with a common downstream reverse gene-specific primer, 5′-TTCTCAGATCTGTCGTTTGCTAA-3′. These reactions isolated a 1.55-kb or 0.28-kb promoter-sequence, immediately upstream of the “ATG” translational start codon of the At5g55380 gene (Appendix A). For the At5g55340 WS gene, a 1.33-kb genomic DNA fragment immediately upstream of the “ATG” translational start codon of the gene was PCR-amplified with the forward gene-specific sequence, 5′-TGATGATTTGGGAAGAGAACTA-3′ and the reverse gene-specific sequence, 5′-CTCTCAGATCTTTGTTTGTGTTG-3′ (Appendix A). The resulting PCR fragments were initially cloned into an entry vector (pDONR221), then into a plant transformation vector pBGWFS7 [58], immediately upstream of promoter-less GFP-GUS gene.

Subcellular localization of the WS genes (At5g55340 or At5g55380) was assessed using transgenic plants, which carried a reporter gene (GFP) fused to the C-terminus of each WS ORF. These constructs were made using the *attB*-enabled strategy described above, by PCR-amplifying a genomic DNA fragment that contained the WS promoter fragment, and the WS ORF to the penultimate translational codon (i.e., excluding the stop codon). For the At5g55380 WS gene, a 2.58-kb fragment was amplified with the gene-specific sequences, 5′-ACCTTGTTCCTCCGCCACT-3′ (forward primer) and 5′-AATGAAGAAGTGAATAACTTGG-3′ (reverse primer). For the At5g55340 WS gene an equivalent 2.33-kb fragment was amplified with the primers containing the gene-specific sequences, 5′-TGATGATTTGGGAAGAGAACTA-3′ (forward primer) and 5′-AATCGCTTAATGAACTCAACG-3′ (reverse primer). Both PCR-products were initially cloned into an entry vector, pDONR221, then into the binary vector, pBGWFS7 (Karimi et al., 2002), upstream of promoter-less GFP-GUS gene (Gateway™ Cloning Technology, Invitrogen, Carlsbad, CA, USA).

### 4.3. Heterologous Expression in Saccharomyces Cerevisiae

In order to improve protein expression in heterologous systems, all WS ORF sequences were codon-optimized for yeast using the GenSmart™ Codon Optimization tool (GenScript, Piscataway, NJ, USA). Chemically synthesized DNAs were obtained from GenScript (Piscataway, NJ, USA), and they were cloned in the plasmid, pUC57. All ORF-coding sequences were subcloned into expression systems using Gateway Technology [59]. Each ORF-coding sequence was PCR-amplified from the pUC57-clones with primers that contained the CACC nucleotide sequence at the beginning of the primer positioned at the 5′-end of each ORF. PCR products were purified and added to TOPO cloning reaction, initially cloning the ORF into a Gateway Entry Vector (Invitrogen, Carlsbad, CA, USA), and subsequently, they were transferred to the Gateway destination vector, pYES-DEST52. 

Plasmid pYES-DEST52 carrying different WS ORFs were transformed into yeast (*S. cerevisiae*) strain H1246 (*MATα*; *are1-Δ::HIS3 are2-Δ::LEU2 dga1-Δ::KanMX4 lro1-Δ::TRP1 ADE2*) [34,60] by electroporation [61]. The yeast strain carrying the plasmid pYES2.1 was used as the negative control in subsequent characterizations. Transformed yeast strains were grown at 30 °C for 24 h in 10 mL of the synthetic dropout medium containing 0.17% (*w*/*v*) yeast nitrogen base, 0.5% (*w*/*v*) ammonium sulfate, 2% (*w*/*v*) glucose, and 0.06% (*w*/*v*) dropout supplement lacking uracil. To induce expression, yeast cells were collected by centrifugation and suspended in 25 mL of the same synthetic dropout medium, except glucose was replaced with 2% (*w*/*v*) raffinose. After 24 h of growth, the expression of the WS ORF was induced by culturing the yeast at 20 °C for 24 h in 25 mL of the same synthetic dropout medium, but this time containing 2% (*w*/*v*) galactose and 1% (*w*/*v*) raffinose as the carbon source. In some experiments, this induction medium was also supplemented with different fatty acids or different alcohols as potential substrates for the expressed WS enzymes. At set times after induction, cells were harvested by centrifugation and analyzed for protein expression or accumulation of wax esters. In the substrate-feeding experiments, stock solutions of 10 mM fatty acids and/or 10 mM fatty alcohols (dissolved in ethanol or DMSO) were diluted 1:200 into the culture medium, to a final concentration of 50 µM. In these experiments 10 mL of culture was withdrawn, the cells were collected by centrifugation, and lyophilized for subsequent protein and lipid extraction.

### 4.4. Transgenic Plant Materials

*Arabidopsis thaliana*, ecotype Columbia (Col-0) was used for all transgenic experiments. Seeds were planted on MS Agar medium in Petri dishes and stratified at 6 °C for 24 h. Seedlings were grown under continuous white fluorescent lighting at 22 °C. After two weeks, seedlings were transferred to LC1 soil mix (Sungro Horticulture, Agawam, MA, USA). Arabidopsis transgenic plants that were expressing WS sequences were generated by a floral dip method, using *Agrobacterium tumefaciens* strain c58C1 [62]. This strain was individually transformed by electroporation, with the plant binary vectors, pKF or pKF-derivatives that were designed to co-express each WS homolog with the jojoba FAR and jojoba KCS. Siliques, rosette leaves, and seeds from the resulting transgenic Arabidopsis lines were collected for genetic and biochemical analysis. Arabidopsis lines carrying GUS and GFP reporter transgenes were generated by a similar method but using the *A. tumefaciens* strain GV3101. In all cases, either T2 or T3 generation plants were analyzed, and multiple independently transformed lines (6–15 lines) for each construct were recovered.

### 4.5. Reverse Transcription-PCR

RNA was extracted from plant material using RNeasy RNA Extraction Kit (Qiagen, Valencia, CA, USA), and the integrity of the isolated RNA was evaluated by agarose gel electrophoresis. The RNA preparation was treated with DNase I (Invitrogen, Carlsbad, CA, USA) to digest contaminating genomic DNA. RNA samples were then used for first strand cDNA synthesis with oligo (dT)_20_ primers using a SuperScript III First Strand Synthesis System for RT-PCR (Invitrogen, Carlsbad, CA, USA). Expression of transgenes was validated by RT-PCR assays using RNA samples isolated from developing siliques of transgenic T3 generation plants, using primers specific to each WS sequence. The specificity of these primer pairs was tested by using RNA isolated from wild-type Arabidopsis to ensure that they did not show false positive signals due to non-specific priming of the PCR reaction. RNA template was extracted from Arabidopsis siliques at early and middle maturation stages of seed development when the glycinin promoter would be expected to be most active [63]. Control assays that evaluated genomic DNA contamination were duplicate PCR reactions with template RNA that had not been reverse transcribed. The Arabidopsis *Actin-2* mRNA (At3g18780) served as a positive control for all RT-PCR reactions [64].

### 4.6. Extraction of Lipids from Seed Samples

A total of 5 mg of desiccated Arabidopsis seeds were homogenized using a glass homogenizer in 1 mL chloroform and 1 mL methanol [65]. Behenyl dodecanoate ester was added as the internal standard. The extract was transferred to a glass tube with a Teflon lined screw cap, and vigorously mixed for 20 min. After the addition of 1.3 mL of hexane and 0.7 mL diethyl ether, the mixture was vigorously mixed for 20 min. After centrifugation for 5 min at 450× *g*, the pellet was discarded, and the supernatant was collected and evaporated under a stream of nitrogen gas, and the recovered lipid was dissolved in 1 mL chloroform for GC-MS or GC-FID analysis.

### 4.7. Extraction of Lipids from Yeast Samples

The yeast lipids were extracted using a modification of a method described previously [66]. Prior to extraction, a solution of 5 mg/mL behenyl dodecanoate was added as an internal standard at a rate of 10 µL per 100 mg dry cell-pellet. Lipids were extracted by adding a 1 mL of near-boiling methanol (60 °C) to the dried yeast cells and the sample was vortexed and immediately incubated at 60 °C for 1 min. After cooling to room temperature, 500 µL of chloroform was added to the sample, and mixed by vortexing for 30 s. Following the addition of 400 µL of H_2_O, the sample was further vortexed for 5 min. An additional 500 µL of chloroform was added, and again vortexed for 30 s. After a final addition of 500 µL of H_2_O and vortexing for 30 s, and two phases were separated by centrifugation at 5000× *g* for 4 min, and the lower chloroform layer was recovered, filtered through a 0.45 µm PTFE filter, and evaporated to dryness using a stream of nitrogen gas.

### 4.8. Thin-Layer Chromatography

Lipid extracts were dissolved in hexane, and the lipid solution was spotted on silica gel plates (Sigma-Aldrich, St. Louis, MO, USA) and developed with the solvent hexane:diethyl ether:acetic acid (90:7.5:1, *v*/*v*) [26]. TLC reference standards (alkane (C_22_H_46_); sterol ester (cholesterol dodecanoate); wax ester (behenyl dodecanoate, C_34_H_68_O_2_); triacylglycerols (canola oil); fatty acid (palmitic acid); fatty alcohol (octadecanol); and sterol (ergosterol) were dissolved in hexane at a concentration of 0.5 mg/mL. The TLC plate was air dried after development, and the plate was sprayed using a glass sprayer, with 0.05 mg/mL Primuline solution (dissolved in acetone/water, 80/20 *v*/*v*) [67]. Lipid spots on the TLC plates were visualized under UV illumination.

### 4.9. GC-MS Analysis

Lipid samples were silylated for GC analysis [68], and a 1-µL aliquot was injected into an Agilent Technologies Model 6890 Gas Chromatograph equipped with an Agilent 19091S-433 column (30.0 m × 250 µm × 0.25 µm). MS-analysis was conducted by coupling the column eluent to a Model 5973 Mass Selective Detector capable of electrical ionization (EI). The chromatography temperature program initially started at 50 °C for 1 min, then raised at a rate of 25 °C per min to 200 °C, held at 200 °C for 2 min, and again raised at a rate of 10 °C per min to 280 °C and maintained at that temperature for 2 min, then raised at 20 °C per min to 320 °C, where it was maintained for 20 min.

### 4.10. Western Blot Analysis

Protein extracts were prepared from yeast cell pellets using Invitrogen Standard Protocol (Invitrogen, Carlsbad, CA, USA). Protein samples were resolved by SDS–PAGE, and the proteins were transferred to a PVDF (polyvinylidene difluoride) membrane (GE Healthcare Bio-Science KK, Piscataway, NJ, USA). Primary antibody was anti-His tag monoclonal antibody (BioRad, Hercules, CA, USA). Secondary antibody was goat–anti-mouse Ig conjugated to horseradish peroxidase (ECL). Blots were developed with Pierce Super Signal West Pico Chemiluminescent substrate (Thermo Fisher Scientific, Rockford, IL, USA). Protein signals were detected using the enhanced chemiluminescence method (ECL, Amersham, Piscataway, NJ, USA).

### 4.11. Histological GUS Activity Analysis

Transgenic lines containing promoter-GUS constructs were harvested at defined stages of development. A minimum of 3 plants were harvested for each independently transformed line; plants or organs from the same line were stained together, in the same solution. Plant tissue was stained at room temperature for a period ranging between 3 to 18 h, as previously described [69]. Staining solution consisted of 0.5 mg/mL X-gluc, 0.1% (*v*/*v*) Triton X-100, 0.4% (*v*/*v*) ethanol, 0.4 mM potassium ferricyanide, 0.4 mM potassium ferrocyanide, and 0.47 M potassium phosphate buffer (pH 7.0). Stained tissues were observed and documented using an Olympus stereomicroscope at the Bessey Microscopy Facility (Iowa State University, Ames, IA, USA).

### 4.12. GFP Fluorescence Microscopy

GFP fluorescence was imaged at the Confocal Microscopy Facility (Iowa State University, Ames, IA, USA) using a Leica TCS NT laser scanning microscope. Images were illuminated by an Argon/Krypton laser and fluorescence was monitored at wavelengths of 488 nm/568 nm (Omnichrome, Chino, CA, USA). Digital images were processed with Adobe Photoshop 7.0 (Adobe, San Jose, CA, USA).

## 5. Conclusions

In this study we characterized a class of wax ester producing enzymes, which occur uniquely in Plantae and belong to the broader class of MBOAT-type enzymes. As typified by Arabidopsis, plant genomes encode multiple isozymes of these WSs. Moreover, plants also encode a second class of enzymes, the bifunctional WSD enzymes, which have been demonstrated to be involved in producing the wax ester constitutes of the plant cuticle. Therefore, to better understand this apparent biochemical and genetic redundancy in wax ester biosynthesis, we developed two heterologous expression systems that demonstrated the broad substrate specificity of plant WSs, in that they not only catalyze the assembly of wax esters, but also the ethyl and benzyl esters. Additionally, we developed transgenic systems to evaluate the expression patterns of two WS genes at the tissue and subcellular levels of spatial resolution. These studies, in combination with insights gained from publicly available RNA-Seq transcriptomics data, support the conclusion that WSs could be involved in assembling wax esters of the cuticle and of esters that have additional physiological functionalities.

## Figures and Tables

**Figure 1 metabolites-12-00577-f001:**
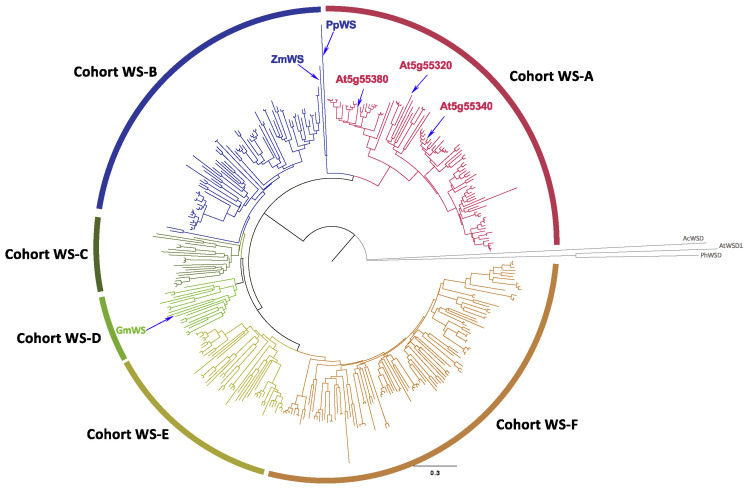
Phylogenetic analysis of MBOAT-type wax synthases. The 375 sequences used in these analyses are identified in Appendix A, and these sequences were identified by sequence homology to the jojoba WS sequence (Accession AAD38041.1) using TBLASTP. The phylogenetic tree was assembled with neighbor joining method, using the experimentally characterized WSD sequences from Arabidopsis (AtWSD1) [26], *A. calcoaceticus* (AcWSD) [24] and *P. hybrida* (PhWSD) [31] as the outgroup, and the statistical rigor of the tree was tested with a bootstrap calculation (*n* = 1000).

**Figure 2 metabolites-12-00577-f002:**
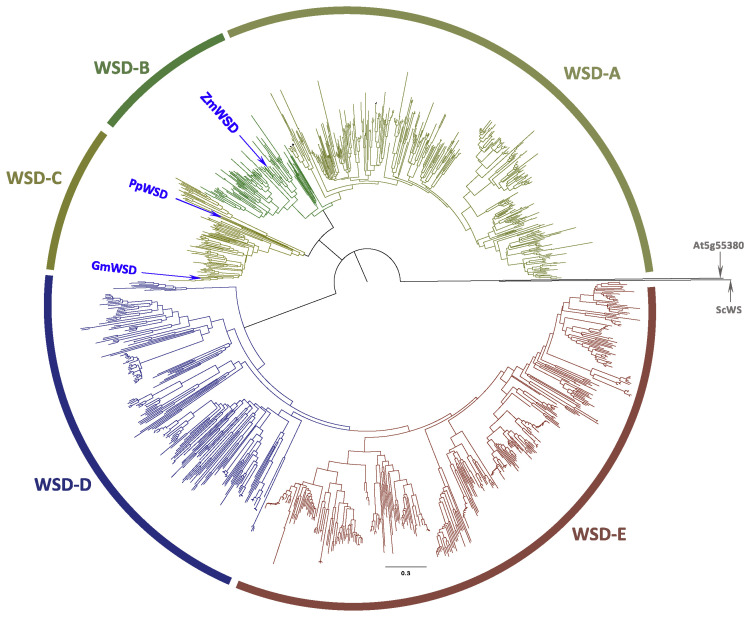
Phylogenetic analysis of bifunctional WSD enzymes. The 1054 sequences used in these analyses are identified in Appendix A, and these sequences were identified by sequence homology to the Arabidopsis (AtWSD1) [26] and *A. calcoaceticus* (AcWSD) [24] sequences using BLASTP. The phylogenetic tree was assembled with neighbor joining method, using the experimentally characterized WS sequences from jojoba WS (ScWS) [22] and the Arabidopsis WS encoded by At5g55380 as the outgroup, and the statistical rigor of the tree was tested with a bootstrap calculation (*n* = 1000).

**Figure 3 metabolites-12-00577-f003:**
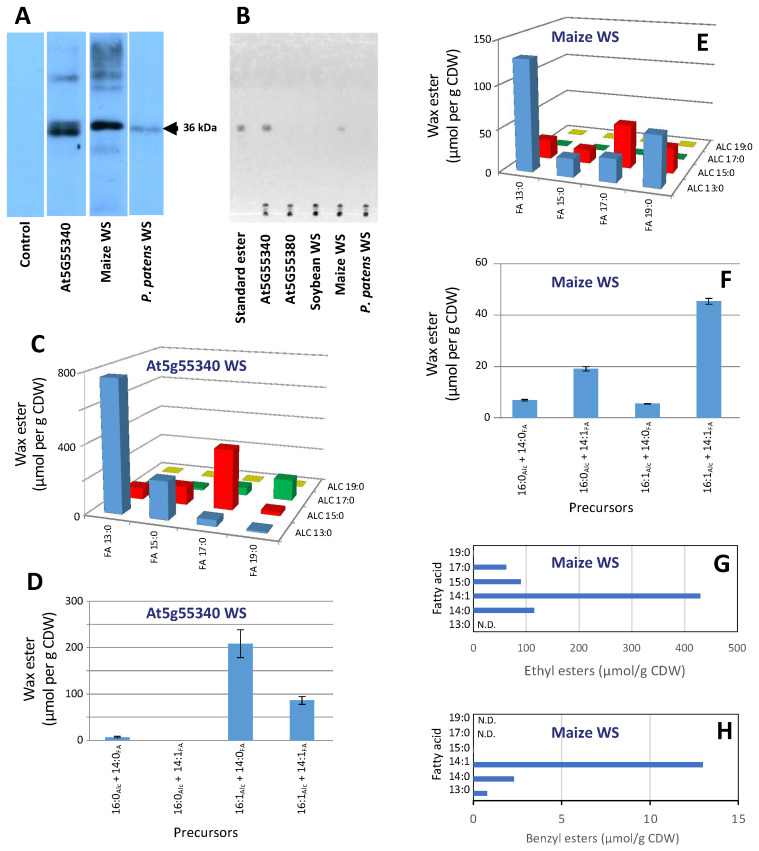
Characterization of WSs expressed in yeast. (**A**) Immunological Western blot detection of the expression of the WSs from Arabidopsis (encoded by At5g55340), maize (encoded by Zm00001d048476), and *P. patents* (encoded by XP_001765175); (**B**) TLC analysis of neutral lipids extracted from yeast strains expressing the indicated WS sequences, which had been grown in media containing the indicated combination of fatty alcohol and fatty acid precursors; (**C**,**D**) wax esters produced by the yeast strain expressing the Arabidopsis WS encoded by At5g55340; (**E**,**F**) wax esters produced by the yeast strain expressing the maize WS encoded by Zm00001d048476; (**G**) ethyl esters produced by the yeast strain expressing the maize WS encoded by Zm00001d048476; (**H**) benzyl esters produced by the yeast strain expressing the maize WS encoded by Zm00001d048476.

**Figure 4 metabolites-12-00577-f004:**
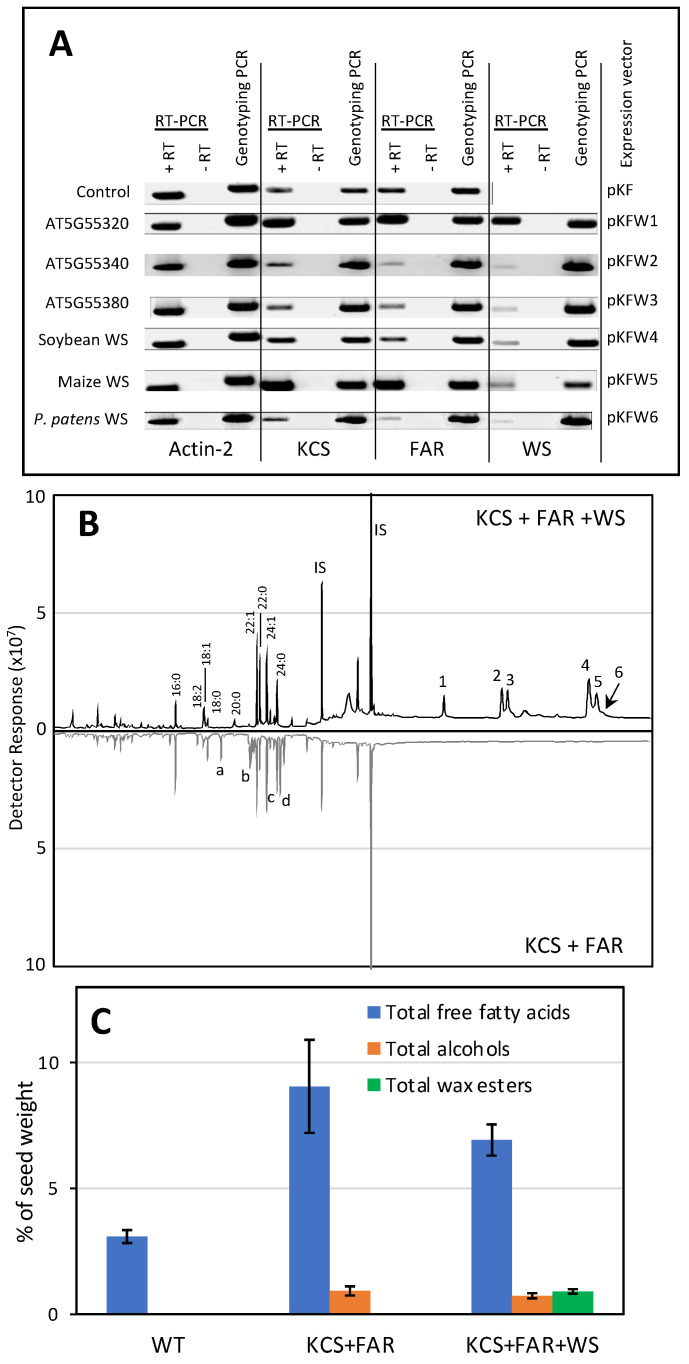
Characterization of WS expressed in Arabidopsis seeds. (**A**) RT-PCR and genotyping PCR of transgenic Arabidopsis plants transformed with the expression vectors pKF (control) and pKFW1-pKFW6. Each of these vectors, except pKF, co-expressed the jojoba KCS, jojoba FAR and each of indicated WS transgenes. RNA isolated from Arabidopsis seeds was used as a template in RT-PCR to confirm the expression of each indicated transgene, and the expression of the Arabidopsis *actin-2* gene was used as a positive control. Control reactions that were not treated with reverse transcriptase (-RT) confirmed that the visualized bands were derived from the mRNA template. In parallel, DNA isolated from the seeds was used to conduct genotyping PCR, which confirmed the presence of each transgene. The identity of each PCR product was confirmed by direct sequencing; (**B**) GC profiles of lipids extracted from Arabidopsis seeds transgenically co-expressing the jojoba KCS and jojoba FAR, with (upper profile) or without (lower profile) the transgenic WS encoded by At5g55380. Identified peaks are: fatty acids (16:0, 18:0, 18:1, 18:2, 18:3, 20:0, 22:0, 22:1, 24:1, 24:1), fatty alcohols (20:1 (a), 22:1 (b), 24:0 (c), 24:1 (d)) and the wax esters: (1) icosenyl hexadecanoate (20:1–16;0); (2) icosenyl octadecadienoate (20:1–18:2); (3) docosenyl hexadecanoate (22:1–16:0); (4) docosenyl octadecaenoate (22:1–18:1); (5) docosenyl octadecenoate (22:1–18:0); (6) tetracosenyl hexadecaenoate (24:1-C16:0); (**C**) total free fatty acids, alcohols, and wax ester extracted from Arabidopsis seeds transgenically co-expressing the jojoba KCS and jojoba FAR, with the transgenic WS encoded by At5g55380. Fatty acids, fatty alcohols, and wax ester content was quantified relative to the peak area of the internal standard, behenyl dodecanoate (21:0–22:0) (peak IS in panel B).

**Figure 5 metabolites-12-00577-f005:**
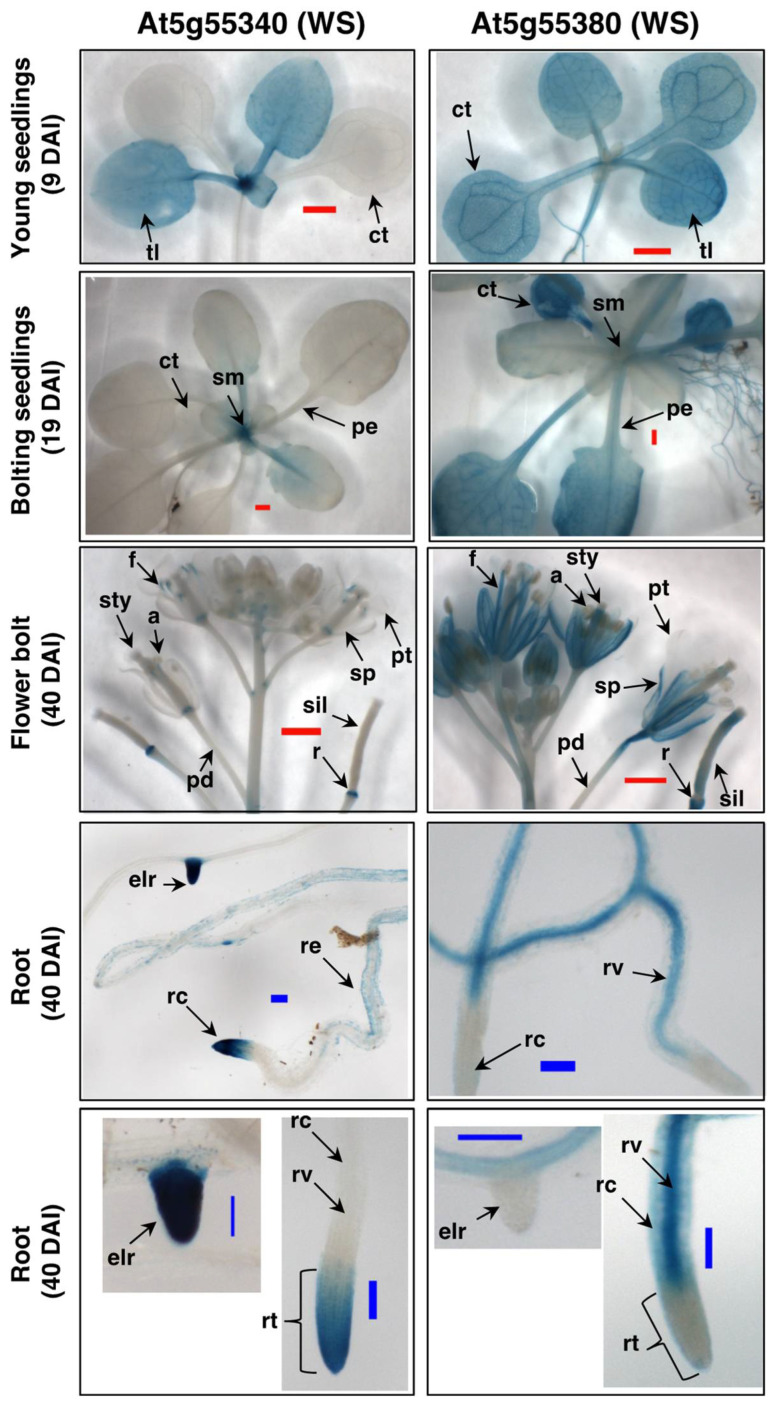
Characterization of WS gene expression with promoter-GUS transgenes. GUS activity in transgenic plants carrying promoter-GUS fusions for the WS genes At5g55340 and At5g55380. Plant organs, and ages of the plants are indicated as days after imbibition (DAI). Identified tissues are: anthers (a); emerging lateral root (elr); filament (f); pedicle (pd); petiole (pe); petal (pt); receptacle (r); root cortex (rc); root cap (rcp); root epidermis (re); root tip (rt); root vasculature (rv); shoot meristem (sm); sepal (sp); stigma (st); style (sty); and true leaf (tl). Scale bars represent: 1 mm (red bar) and 0.1 mm (blue bar).

**Figure 6 metabolites-12-00577-f006:**
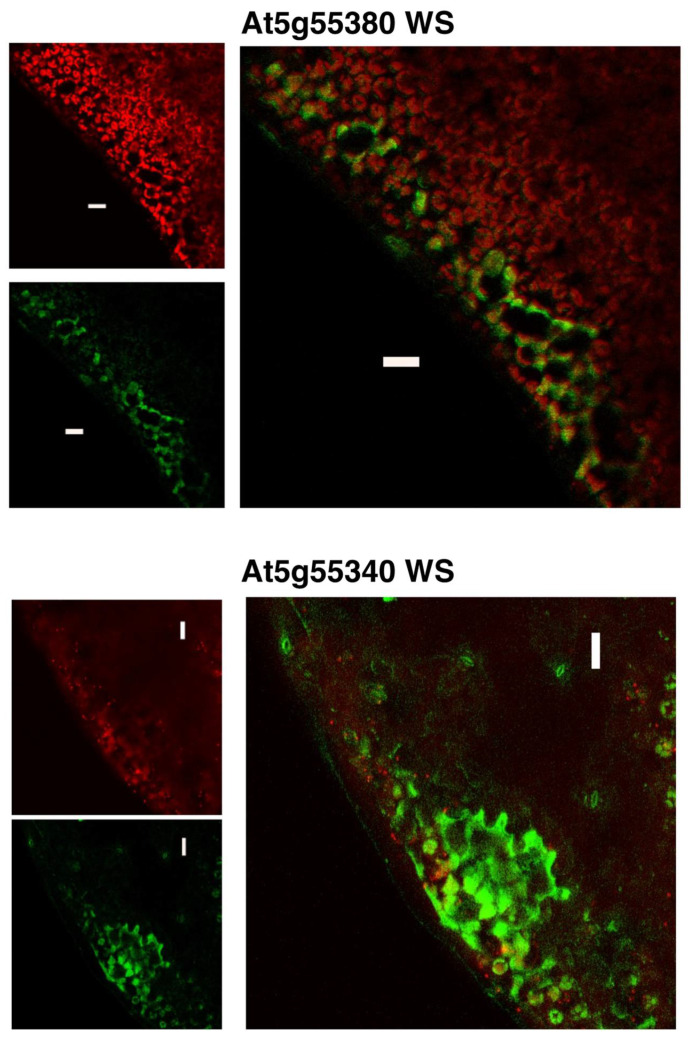
Subcellular localization of WS proteins. GFP fluorescence generated by the transgenic expression of WS-GFP fusions is compared to the auto-fluorescence that visualizes the location of chloroplasts. Images are of the epidermis and subepidermal cell layer of leaf tissue from 15 DAI transgenic plants expressing GFP-fusions of the indicated WS genes.

**Figure 7 metabolites-12-00577-f007:**
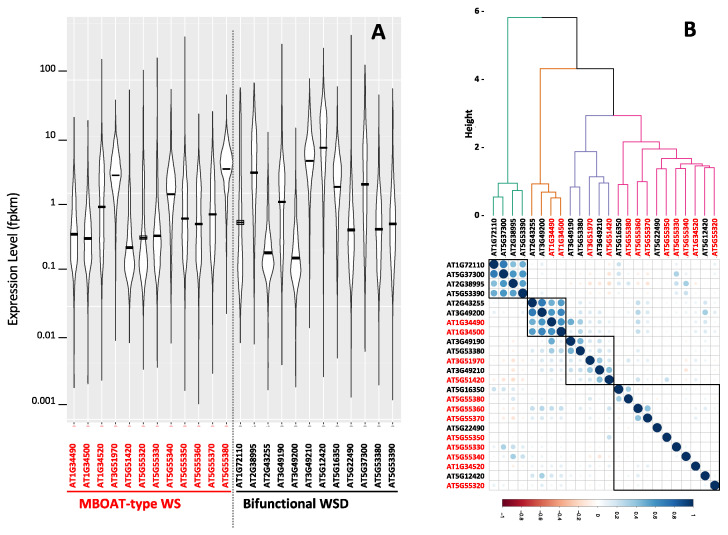
Expression levels and correlations among the Arabidopsis WS and WSD genes. (**A**) Violin plot presentations of the range of expression levels of the 12 WS genes and 11 WSD genes determined from 5200 individual Arabidopsis RNA-Seq experiments downloaded from the NCBI Sequence Read Archive (Appendix A); (**B**) correlation of the expression of each WS and WSD genes with all Arabidopsis genes calculated from the 5200 individual Arabidopsis RNA-Seq experiments downloaded from the NCBI Sequence Read Archive (Appendix A).

**Table 1 metabolites-12-00577-t001:** Arabidopsis was synthase and bifunctional WSD coding genes.

Enzyme Type	Gene ID	Protein Length (aa)	Subcellular Location ^a^		Maximum Expression ^b^		
			PSORT Prediction	SUBA Prediction	Organ	Expression Level (fpkm)	SRA Accession
Wax synthase	At1g34490	337	TM or PM	PM	Root cells (isolated)	22.0	SRR8206657
	At1g34500	341	PM	PM	Leaves (7-d old seedlings)	19.3	SRR8742425
	At1g34520	336	TM or PM	PM	Apical Meristem	233.3	SRR390310
	At3g51970	345	TM	PM	Stigma (non-pollinated)	44.3	ERR2278241
	At5g51420	435	TM or PM	PM	Inflorescence meristem	66.7	SRR5681054
	At5g55320	339	PM or TM	PM	Developing seeds (3-d after pollination)	149.7	SRR1232482
	At5g55330	346	PM or TM	PM	Siliques (1-d after pollination)	246.4	SRR3347475
	At5g55340	333	PM	PM	Developing embryos (globular stage)	68.4	SRR8249028
	At5g55350	345	mitoIM or PM	PM	Stem	578.8	SRR2037335
	At5g55360	342	PM or TM	PM	Seedlings (10-d after germination)	25.0	SRR3707607
	At5g55370	343	PM or TM	PM	Seedlings (7-d after germination)	27.7	SRR4734675
	At5g55380	341	TM or PM	PM	Inflorescence meristem	41.3	ERR1698199
WS/ DGAT	At1g72110	479	ER	Perox	Stem	73.2	SRR4007446
	At2g38995	487	ER	PM	Siliques	88.7	SRR3347480
	At3g49190	522	PM	Cysk	Leaves (7-d old seedlings)	11.9	SRR8742425
	At3g49200	507	PM	Cysk	Root cells (isolated)	430.2	SRR8206660
	At3g49210	518	ER	Cysk	Leaves (7-d old seedlings)	14.2	SRR8742425
	At5g12420	480	PM	Cysk	Roots (6-d old seedlings)	367.6	SRR5195558
	At5g16350	488	Cytpl	Perox	Hypocotyl	76.4	SRR6312333
	At5g22490	482	ER	Cysk	Stem	619.0	SRR2037351
	At5g37300	481	Perox	Cysk or Perox	Flowers (6-week old plants)	184.2	SRR6179906
	At5g53380	483	Perox	Cysk	Root cells (isolated)	54.8	SRR8206659
	At5g53390	485	PM	Cysk	Seedling	70.4	ERR1876169

^a^ Subcellular localization predictions obtained with PSORT (http://psort1.hgc.jp/form.html, accessed on 15 March 2021) and SUBA (https://suba.live, accessed on 15 March 2021). Abbreviations: Cytoplasm (Cytpl); cytoskeleton (Cysk) endoplasmic reticulum (ER); mitochondrial inner membrane (mitoIM); peroxisomes (Perox); plasma membrane (PM); thylakoid membrane (TM). ^b^ Arabidopsis organ/tissue sample that show the maximum expression of each WS and WSD gene as determined by RNA-Seq experiments recovered from the Sequence Read Archive (SRA) maintained by NCBI (www.ncbi.nlm.nih.gov/sra, accessed on 10 December 2020).

## Data Availability

The study design information, GC–MS data, data processing, and analyses are reported and incorporated into the manuscript and Appendix A, and all data are available on request.

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
