# Peer review of "Heterologous Expression and Characterization of Plant Wax Ester Producing Enzymes"

_metabolites, 2022, doi:10.3390/metabo12070577_

Round 1

Reviewer 1 Report

In the Ms “Heterologous Expression and Characterization of Plant Wax Ester Producing Enzymes” Authors have done an inclusive analysis including in-Silico to functional characterization. They did phylogenetic analysis of 12 WS and 11 WSD isozymes that occur in Arabidopsis, and established two in vivo heterologous  expression systems, in the yeast and in Arabidopsis seeds to investigate the catalytic abilities of the WS enzymes. These two refactored wax assembly chassis were used to demonstrate that WS isozymes show distinct differences in the types of esters that can be assembled. They had also determined the cellular and subcellular localization of two Arabidopsis WS isozymes. Additionally, using publicly available Arabidopsis transcriptomics data they identified the coexpression modules of the 12 Arabidopsis WS coding genes.

This is a complete piece of work. The Ms is also very nicely written. It can be directly accepted for publication.

Author Response

We thank the reviewer for the supportive comments.

Reviewer 2 Report

In this work, we defined a class of wax ester-generating enzymes that belong to the larger family of MBOAT-type enzymes that only exist in Plantae. These enzymes are found nowhere else in nature. Because the work has excellent writing, methodology and appropriate data, it is possible that it will be approved for publication.

Author Response

We thank the reviewer for the supportive comments

Reviewer 3 Report

The article "Heterologous Expression and Characterization of Plant Wax Ester Producing Enzymes" is well thought out and contains interesting information.
Pay attention to the spaces in the text!
The abbreviations WS and WSD are only explained in the abstract, it would be good to repeat this in the introduction.

Also in the heading of Table 1 only abbreviations are used, please use full words.
Is it necessary to use the term promiscuity in the conclusion?
The references are well cited except reference 51, correct it.

Author Response

We thank the reviewer for the constructive comments.  We have made the following revisions, in response to the reviewer's suggestions:

1) Defined both WS and WSD in the Introduction

2) Spelled out names of enzymes in header of Table 1

3) rephrased the conclusion to avoid the use of the word "promiscuity"

4) corrected citation #51